# Acute cerebellar knockdown of *Sgce* reproduces salient features of myoclonus-dystonia (DYT11) in mice

Samantha Washburn, Rachel Fremont, Maria Camila Moreno-Escobar, Chantal Angueyra, Kamran Khodakhah*

Dominick P. Purpura Department of Neuroscience, Albert Einstein College of Medicine, New York, United States

**Abstract** Myoclonus dystonia (DYT11) is a movement disorder caused by loss-of-function mutations in *SGCE* and characterized by involuntary jerking and dystonia that frequently improve after drinking alcohol. Existing transgenic mouse models of DYT11 exhibit only mild motor symptoms, possibly due to rodent-specific developmental compensation mechanisms, which have limited the study of neural mechanisms underlying DYT11. To circumvent potential compensation, we used short hairpin RNA (shRNA) to acutely knock down S*gce* in the adult mouse and found that this approach produced dystonia and repetitive, myoclonic-like, jerking movements in mice that improved after administration of ethanol. Acute knockdown of *Sgce* in the cerebellum, but not the basal ganglia, produced motor symptoms, likely due to aberrant cerebellar activity. The acute knockdown model described here reproduces the salient features of DYT11 and provides a platform to study the mechanisms underlying symptoms of the disorder, and to explore potential therapeutic options.

## Introduction

Dystonia is a hyperkinetic movement disorder characterized by involuntary co-contraction of agonist and antagonist muscles resulting in abnormal, repetitive movements and pulled or twisted postures that cause varying degrees of disability and pain (*Albanese et al., 2013*). The prevalence of dystonia in the general population has been notoriously difficult to determine precisely, due to different methodologies for case classification, but a recent meta-analysis estimated that primary dystonia occurs at a rate of approximately 16 per 100,000 (*Steeves et al., 2012*). This is likely an underestimate, as cases frequently go undiagnosed and dystonia can also be a secondary symptom in other motor disorders. While dystonia most often occurs sporadically, mutations in several genes have been identified and linked with specific forms of dystonia (*Charlesworth et al., 2013*). Myoclonus-dystonia (DYT11) is an inherited form of dystonia caused by loss-of-function mutations in the *SGCE* gene, which encodes the protein epsilon sarcoglycan (ε-SG) (*Zimprich et al., 2001*). The vast majority of DYT11 patients with *SGCE* mutations inherit the disorder from the male parent, with markedly reduced penetrance in the offspring of female patients (*Asmus et al., 2002*; *Zimprich et al., 2001*). The disorder is maternally inherited in only 5–10% of cases. This pattern of dominant, primarily paternal inheritance is consistent with imprinting, or epigenetic silencing, of the maternal allele. Indeed, in DNA and RNA samples from human blood *SGCE* is maternally imprinted, but the imprinting pattern in the brain is unknown (*Grabowski et al., 2003*).

DYT11 is characterized primarily by involuntary jerking of the arms, neck, and trunk (myoclonus) and oftentimes mild to moderate dystonia that worsens with stress (*Kyllerman et al., 1990*). Motor symptoms usually appear in childhood or adolescence (*Nardocci et al., 2008*; *Raymond and Ozelius, 1993*; *Raymond et al., 2008*) and are frequently accompanied by psychiatric symptoms,

*For correspondence:
k.khodakhah@einstein.yu.edu

**Competing interests:** The authors declare that no competing interests exist.

including anxiety, panic attacks, and obsessive compulsive disorder (*Peall et al., 2011*; *Weissbach et al., 2013*). DYT11 is known as alcohol-responsive dystonia given that approximately half of patients with DYT11 report improvement of motor symptoms after consuming alcohol. Strikingly, for these patients, alcohol tends to provide more therapeutic relief than any available pharmacological interventions, which have frequently been ineffective or poorly-tolerated. Unfortunately, the addictive and neurodegenerative consequences of chronic alcohol use preclude its use as a therapeutic option. Although patients generally live an active life of normal span, the motor and psychiatric symptoms of this disorder can cause a great deal of physical and psychosocial distress, significantly impacting the patients' quality of life.

A limiting factor in the development of more effective treatment strategies for patients with DYT11 is the lack of understanding of the neural basis of the disorder. While the genetic cause of DYT11 has been known for 10 years, the function of ε-SG protein, particularly in the nervous system, and how loss-of-function of ε-SG leads to motor symptoms remains elusive. This issue is not specific to DYT11, but rather a common problem in understanding genetic dystonias. While mutations in the genes encoding α-, β-, γ-, and δ-sarcoglycans cause different forms of muscular dystrophies, no signs or symptoms of muscle disease have been detected in DYT11 patients with mutations in *SGCE* (*Hjermind et al., 2008*). Moreover, electrophysiological and fMRI studies have cited a subcortical origin of the disorder (*Marelli et al., 2008*; *Nitschke et al., 2006*; *Roze et al., 2008*). Generally, dystonia has been viewed as a disorder of the basal ganglia. Indeed, in patients with DYT11, changes have been observed in the internal global pallidus (GPi), an output structure of the basal ganglia. In particular, studies have shown that gray matter volume of the GPi (*Beukers et al., 2011*) and firing patterns of individual neurons in the GPi (*Welter et al., 2015*) correlates with the severity of motor symptoms in patients. Moreover, deep brain stimulation of the GPi is effective for most DYT11 patients, particularly those with *SGCE* mutations (*Azoulay-Zyss et al., 2011*; *Fernández-Pajarín et al., 2016*; *Gruber et al., 2010*; *Kosutzka et al., 2019*; *Rocha et al., 2016*; *Rughani and Lozano, 2013*).

On the other hand, recent studies have shown that dystonia is associated with changes in the activity, structure, and connections of the cerebellum (*Draganski et al., 2003*; *Dresel et al., 2014*; *Hendrix and Vitek, 2012*; *Jinnah and Hess, 2006*; *LeDoux et al., 1993*; *LeDoux et al., 1995*; *Prell et al., 2013*; *Shakkottai, 2014*; *Song et al., 2014*). In DYT11 patients, there is evidence from fMRI (*Beukers et al., 2010*; *Nitschke et al., 2006*; *van der Salm et al., 2013*), and PET (*Carbon et al., 2013*) for increased cerebellar activity. There are also diffusion tensor imaging data to suggest that changes occur in the white matter tracts that connect the cerebellum to the thalamus and other parts of the brain (*van der Meer et al., 2012*). Furthermore, studies on the expression of *SGCE* and in the human brain and *Sgce*, the mouse homolog of *SGCE*, in the rodent brain have suggested that a brain-specific isoform, which might be responsible for the purely neurophysiological phenotype of DYT11, is highly expressed in the principle computational and output neurons of the cerebellum, the Purkinje cells and deep cerebellar nuclei (DCN) neurons, respectively, while it is expressed at lower levels in the different nuclei of the basal ganglia (*Ritz et al., 2011*; *Xiao et al., 2017*; *Yokoi et al., 2005*). Lastly, the amelioration of symptoms after ethanol ingestion reported by a number of patients with DYT11 points to a role for the cerebellum in this disorder, as the cerebellum is exquisitely sensitive to alcohol. Several labs have demonstrated that alcohol modulates Purkinje cell spiking activity (*Chu, 1983*; *Rogers et al., 1980*; *Sinclair et al., 1980*), increases inhibitory synaptic transmission in the cerebellum (*Carta et al., 2004*), and alters synaptic plasticity at synapses onto Purkinje cells (*Belmeguenai et al., 2008*; *He et al., 2013*).

The goal of this study was to establish a mouse model that would enable the investigation of the neural correlates of motor symptoms in DYT11. Previous genetic knockout models have had subtle effects on motor behavior. Mice with a ubiquitous genetic knockout of *Sgce* exhibited full body jerks and had difficulty learning in a balance beam task (*Yokoi et al., 2006*). Conditional knockout of *Sgce* in Purkinje cells of the cerebellum (*Yokoi et al., 2012a*) or specifically in the striatum (*Yokoi et al., 2012b*), the input nucleus of the basal ganglia, produced milder effects. Importantly, none of these mice exhibited dystonia or alcohol-responsiveness, both prominent features of DYT11. Unfortunately, genetic models of hereditary dystonia have often failed to recapitulate many of the key motor symptoms of the disorder. One possibility for this phenomenon is compensation for the gene that has been knocked out during brain development in rodents (*Fremont and Khodakhah, 2012*).

In recent years, our laboratory has successfully generated symptomatic mouse models of dystonia using an acute knockdown approach to overcome potential developmental compensation (*Calderon et al., 2011*; *Fremont et al., 2014*; *Fremont et al., 2017*; *Fremont et al., 2015*). Here, we used short hairpin ribonucleic acid (shRNA) to knockdown *Scge* mRNA selectively in the cerebellum or basal ganglia of adult mice. We found that knockdown of *Sgce* in the cerebellum, but not the basal ganglia, produced overt motor symptoms, including dystonia. Consistent with what is seen in patients, administration of ethanol improved motor symptoms in *Sgce* knockdown mice but had no effect on motor symptoms of mice with knockdown of the gene that causes primary torsion dystonia (DYT1). We further found that, compared to mice injected with a control shRNA, both Purkinje cells and DCN neurons of the cerebellum fire aberrantly in *Sgce* knockdown mice.

## Results

We hypothesized that developmental compensation for genetic knockout of *Sgce* could account for the mild motor symptoms observed in previous mouse models of DYT11. To address this issue, we employed an acute, shRNA-mediated knockdown strategy to generate a symptomatic mouse model of DYT11. This strategy was successful in two previous studies of rapid onset dystonia-parkinsonism (DYT12) (*Fremont et al., 2015*) and primary torsion dystonia (DYT1) (*Fremont and Khodakhah, 2012*; *Fremont et al., 2017*).

A wealth of evidence suggests that the cerebellum may be particularly important in the pathophysiology of DYT11 (*Beukers et al., 2010*; *Carbon et al., 2013*; *Ritz et al., 2011*; *van der Meer et al., 2012*; *van der Salm et al., 2013*). To examine the contribution of the cerebellum to the symptoms observed in DYT11 patients, we injected AAV9 virus encoding shRNA against *Sgce* mRNA and a GFP reporter (AAV-SGCEshRNA-GFP) into the cerebellum of adult mice (*Figure 1A*). We found that injection of AAV-SGCEshRNA-GFP resulted in expression of the construct throughout the cerebellum without causing any gross morphological abnormalities (*Figure 1B*), as well as significant knockdown of *Sgce* mRNA as determined by qRT-PCR (*Figure 1C*, Mann-Whitney Test, WT vs. NT CB: p=0.7922; NT CB vs. *Sgce* KD CB 1: p=0.0079; NT CB vs. *Sgce* KD CB 2: p=0.0159; $N_{WT}$ = 6, $N_{NT\ CB}$ = 5, $N_{Sgce\ KD\ CB\ 1}$ = 5, $N_{Sgce\ KD\ CB\ 2}$ = 4). To control for shRNA-specific effects, a group of mice were injected with shRNA that does not target any gene in the genome (AAV-shNT-GFP, NT CB). To control for off-target effects of the shRNA, we used two sequences of shRNA that target separate regions of *Sgce* (*Sgce* KD CB one and *Sgce* KD CB 2). Mice were injected with either of two shRNA sequences against *Sgce* into the cerebellum. Their behavior in the open field was assessed by four scorers blind to the condition of the animal using a previously published dystonia scale, where a score greater than or equal to two indicates dystonia (*Calderon et al., 2011*; *Fremont et al., 2014*; *Fremont et al., 2017*; *Fremont et al., 2015*). *Sgce* KD CB mice had a time-dependent induction of dystonia, indicated by a dystonia score of 2 or above (*Video 1*, *Figure 1Di*, *Sgce* KD CB 1: N = 39; *Sgce* KD CB 2: N = 40). Abnormal movements and postures were observed throughout the body of the animal, although the most severe dystonic postures were observed in the hind limbs and tails (*Figure 1D,ii*). In contrast, mice injected with AAV-shNT-GFP did not develop symptoms (*Video 1*, *Figure 1Di*, NT CB: N = 16). The development of dystonic symptoms in *Sgce* KD CB mice is time-dependent because the transfection of the virus, expression of the shRNA, and knockdown of ε-SG protein increases over time. This provides the unique opportunity for within-animal control to make sure that the surgery itself did not cause motor symptoms. Accordingly, the dystonia scores for *Sgce* KD CB 1 and *Sgce* KD CB 2 mice for time points of 2 weeks or more after injection were significantly different from the dystonia scores of the same animals at <1 week (*Figure 1Di*, p<0.01, Wilcoxon matched-pairs signed rank test). Consistent with these data, the dystonia scores of *Sgce* KD CB 1 and *Sgce* KD CB 2 mice at <1 week after injection were not significantly different from NT CB mice at the same time point (*Figure 1Di*, p=0.81 and p=0.97, respectively, t-test, Holm-Sidak method). The dystonia scores of *Sgce* KD CB 1 and *Sgce* KD CB 2 were not significantly different (p>0.05, t-test, Holm-Sidak method).

Approximately 75% (29/39) of *Sgce* KD CB 1 mice and 60% (23/40) of and *Sgce* KD CB 2 mice developed dystonia. To explore whether the absence of symptoms is due to inefficient knockdown of *Sgce* or whether it is due to penetrance of the phenotype, we plotted the extent of *Sgce* knockdown, determined by qRT-PCR, against the dystonia score recorded in a subset of animals (*Figure 1E*). Based on these data, there appears to be a correlation between the severity of

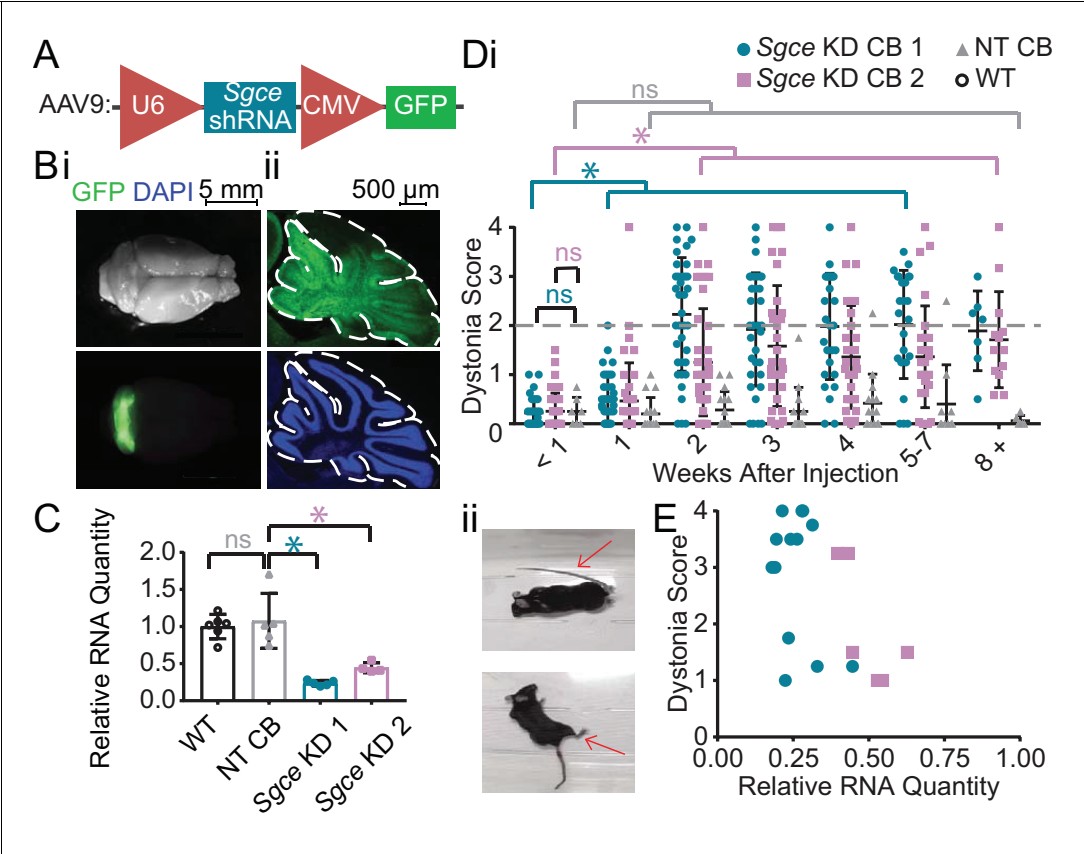

**Figure 1.** shRNA-mediated knockdown of *Sgce* in the cerebellum causes dystonia. (**A**) Schematic of AAV-shSGCE-GFP construct. (**B**) Images of the whole brain (i) and sagittal cerebellar section (ii) from an *Sgce* KD CB mouse. (**C**) Quantification of qRT-PCR confirms that *Sgce* RNA is reduced in vivo. (Mann-Whitney Test, WT vs. NT CB: p=0.7922; NT CB vs. *Sgce* KD CB 1: p=0.0079; NT CB vs. *Sgce* KD CB 2: p=0.0159; $N_{WT}$ = 6, $N_{NT\ CB}$ = 5, $N_{Sgce\ KD\ CB\ 1}$ = 5, $N_{Sgce\ KD\ CB\ 2}$ = 4). (**D,i**) Injection of *Sgce* KD 1 and 2 into the cerebellum produced dystonia, while injection of NT did not. (*Sgce* KD CB 1: N = 39; *Sgce* KD CB 2: N = 40; NT CB: N = 16). Dystonia was measured on a previously published dystonia scale by four scorers blind to the condition of the animal. A score greater than or equal to two indicates dystonia. The dystonia scores for *Sgce* KD CB 1 and *Sgce* KD CB 2 mice for time points of 2 weeks or more after injection were significantly different from the dystonia scores of the same animals at <1 week (Wilcoxon matched-pairs signed rank test, p<0.01). The dystonia scores of *Sgce* KD CB 1 and *Sgce* KD CB 2 mice at <1 week after injection were not significantly different from NT CB mice at the same time point (t-test, Holm-Sidak method, p=0.81 and p=0.97, respectively). (**ii**) Example dystonic postures exhibited by *Sgce* KD CB mice. (**E**) Scatter plot of RNA levels normalized to the mean of WT, determined by qRT-PCR, plotted against the Dystonia Score observed in a subset of animals injected with varying concentrations of shRNA (WT: N = 5, NT: N = 5, *Sgce* KD CB 1: N = 13, *Sgce* KD CB 2: N = 7).

symptoms and knockdown of *Sgce* RNA, consistent with previous data in an shRNA-mediated mouse model of DYT1 (*Fremont et al., 2017*). However, we found that while 5/20 *Sgce* KD CB animals had more than 50% knockdown of *Sgce* RNA compared to WT, they had a dystonia score of less than 2. It is thus possible that, while the extent of *Sgce* knockdown explains some variability observed in the behavioral data, incomplete penetrance of the phenotype may also contribute to the variability of the symptoms observed in *Sgce* KD CB mice.

Upon closer examination of *Sgce* KD CB mice, we noticed that in addition to dystonia, these mice exhibited a range of motor symptoms that we had not observed in some of the other mouse models of movement disorders previously characterized in our laboratory (*Video 2*). *Sgce* KD CB mice had difficulty ambulating normally, showed unsteady or mildly ataxic gate, and exhibited myoclonic-like jerking movements, in addition to overt dystonia. Furthermore, when suspended by the tail, *Sgce* KD CB mice would spin vigorously and, often, continuously. In order to capture the range of motor symptoms observed in *Sgce* KD CB mice more accurately, we generated two new scales to measure the characteristic symptoms seen in this model. The Disability Scale (*Table 1*) took into consideration the frequency of sustained dystonic-like postures and repetitive movements, as well as the level of motor impairment. The Spinning Scale (*Table 2*) was generated to measure the frequency and

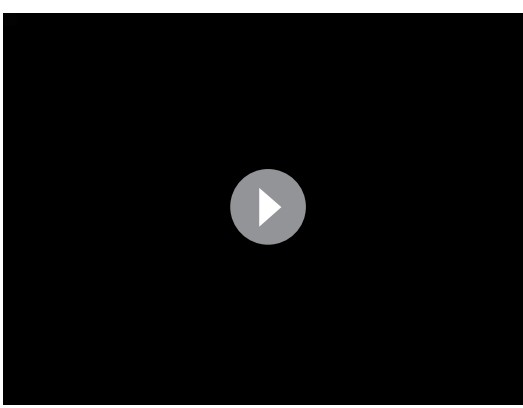

**Video 1.** *Sgce* KD CB 1 and sgce KD CB 2 mice have dystonia, as evidenced by a dystonia score of greater than or equal to two in the open field, while NT CB mice do not develop dystonia (p<0.01, Wilcoxon matched-pairs signed rank test, *Sgce* KD CB 1: N = 39; *Sgce* KD CB 2: N = 40; NT CB: N = 16).
https://elifesciences.org/articles/52101#video1

duration of spinning during a tail suspension test, a symptom unique to *Sgce* KD CB mice compared to our other models of dystonia. The average Spinning Score for a subset of *Sgce* KD CB 1 and *Sgce* KD CB 2 was 2.22 ± 0.77 and 2.00 ± 0.88, respectively (Mean ± S.D., N = 13 and 7). The average Disability Score was 3.02 ± 0.76 and 2.89 ± 0.79 (Mean ± S.D., N = 14 and 8), respectively.

To examine the contribution of the basal ganglia to symptoms in DYT11, we injected shRNA into the striatum and GPi of wild-type mice (*Sgce* KD BG 1 and 2, *Figure 2A*). Mice injected with AAV-SGCEshRNA-GFP into the basal ganglia developed mildly abnormal motor symptoms, but they did not develop dystonia (*Figure 2B*, *Video 3*, *Sgce* KD BG 1: N = 4, *Sgce* KD BG 2: N = 14). In 5 out of 47 observations, animals scored two or above on the Dystonia Scale. Unlike the symptoms observed in *Sgce* KD CB mice, which occurred multiple times in the same recording period and persisted over time, these postures did not occur more than once in the same mouse, suggesting that they might have been false positive identification of dystonic-like postures in those incidences. There were no significant differences in the motor symptoms of *Sgce* KD BG 1 and *Sgce* KD BG 2 mice; consequently, only a small subset of mice was injected with *Sgce* KD BG 1 to confirm the observations made in a larger cohort of *Sgce* KD BG 2 mice. Motor abnormalities were not observed in mice injected with AAV-shNT-GFP into the basal ganglia (*Figure 2B*, *Video 3*, NT BG, N = 9).

As discussed, although *Sgce* KD BG mice did not develop dystonia, they exhibited abnormal motor activity and these were indicated as scores of 1 and 2 on the Dystonia Scale. These higher scores were statistically significant at the three time points examined. Closer examination of the motor behavior of the mice showed that, in many of them, their activity in the periphery of the open field arena was increased; several *Sgce* KD BG mice appeared to travel around the periphery and avoid the center of the open field chamber more than either WT or NT BG mice (*Figure 2C*). In addition to ambulating more on the periphery, these *Sgce* KD BG animals also appeared to spend less time in the center of the open field chamber, compared to the periphery (*Figure 2—figure supplement 1A*). While the tracks of many of the animals were clearly qualitatively different, there was considerable variability as to how the tracks looked in each animal, and as an average, no statistically significant differences were detected in the ratio of average time spent in the center vs. the periphery (*Figure 2—figure supplement 1B*). The individual variability from mouse to mouse could be either a function of the extent of knock down of the protein in the BG, or the extent of coverage of the BG by the injected virus. Our ability to detect statistically significant differences in this analysis was also likely limited by the size of the open field chamber used in our experiments. Similarly, total distance traveled was not statistically different among the groups of mice (*Figure 2—figure supplement 1C*). *Sgce* KD CB mice did not show

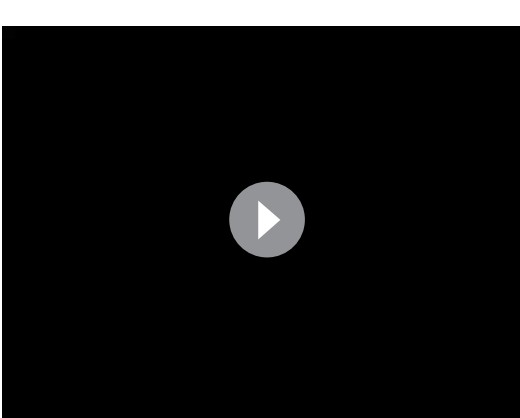

**Video 2.** In addition to dystonia, *Sgce* KD CB mice exhibit myoclonic-like movements in the open field and spin when suspended by the tail.
https://elifesciences.org/articles/52101#video2

**Table 1.** Disability scale for assessing motor impairment in *Sgce* knockdown mouse model of DYT11.

| | |
|---|---|
| 0 | Normal – animal has no observable motor deficit |
| 1 | Slight motor disability – animal may exhibit unsteady gait or uncoordinated movement, but does not have any repetitive movements or sustained postures |
| 2 | Mild motor disability – ambulation is mildly impaired; animal exhibits wide stance, very unsteady gait, back-walking, and/or occasional repetitive movements or sustained postures |
| 3 | Moderate motor disability – ambulation is moderately impaired; animal does not properly ambulate, dragging itself on its side, and may exhibit frequent sustained dystonic-like postures or repetitive movements |
| 4 | Severe motor disability – animal is unable to properly ambulate for most of the duration of the observation, exhibits frequent repetitive movements, and/or sustained dystonic-like postures |

the same qualitative preference for the periphery and avoidance of the center observed in *Sgce* KD BG mice (*Figure 2—figure supplement 2*).

A prominent feature of DYT11 that has not yet been reported in any mouse model is the ability of alcohol to lessen the severity of motor symptoms. To examine whether alcohol could improve the motor symptoms observed in *Sgce* KD CB mice, and further validate this new model, we examined the motor behavior of dystonic (dystonia score ≥2) *Sgce* KD CB mice in the open field before and after a subcutaneous injection of ethanol (2 g/kg) or physiological saline. To capture the full range of motor symptoms that ethanol might affect, we scored the motor behavior of these mice on all three scales. Consistent with what has been reported by patients with DYT11, ethanol relieved motor symptoms in *Sgce* KD CB mice for up to 90 min after injection as measured on both the Disability Scale (*Video 4*, *Figure 3A*, $p<0.0001$, 1way ANOVA, N = 16) and Spinning Scale (*Figure 3B*, $p<0.0001$, 1way ANOVA, N = 19). The reduction in Disability and Spinning Scores were significant at each time point after correcting for multiple comparisons (Holm-Sidak's multiple comparisons test). Similarly, ethanol reduced motor dysfunction as measured on the Dystonia Scale (*Figure 3C*, $p<0.0001$, 1way ANOVA, N = 16). The reduction in dystonia after ethanol was significant at each time point in *Sgce* KD CB mice after correcting for multiple comparisons (Holm-Sidak's multiple comparisons test). To determine whether this effect was specific to *Sgce* KD CB mice, the effect of alcohol was examined in another symptomatic shRNA-mediated mouse model of primary torsion dystonia (DYT1). Alcohol-responsiveness was not observed in this model of DYT1 (*Fremont et al., 2017*), although they exhibit overt dystonia, suggesting that the response to alcohol is specific to ε-SG knockdown (*Video 4*, *Figure 3D*, 0.2391, 1way ANOVA, N = 5). These findings show that acute knockdown of ε-SG in the cerebellum of adult rodents is sufficient to generate ethanol-responsive motor symptoms, including dystonia.

Although ethanol reduced motor symptoms on the Disability, Spinning, and Dystonia Scales in *Sgce* KD CB mice, it did not consistently alter the distance traveled in the open field in each mouse (*Figure 3—figure supplement 1*). While, on average, there was a significant increase in the distance traveled in the open field after EtOH ($p=0.0148$, 1Way ANOVA), this was driven largely by a small number of mice, and the effect was not statistically significant after correcting for multiple comparisons (Dunnett's multiple comparisons test, $p>0.05$ at each time point). In contrast to EtOH, saline had no effect on the Dystonia Score or distance traveled in *Sgce* KD CB mice (*Figure 3—figure supplement 2*, Dystonia Score: $p=0.9517$, Distance Traveled: $p=0.2851$, 1way ANOVA).

Since we successfully generated a symptomatic mouse model of DYT11, we further sought to examine the underlying neural correlates of motor symptoms in DYT11 using electrophysiological techniques. We hypothesized that if irregular activity of the cerebellum contributes to the motor

**Table 2.** Spinning scale for assessing abnormal motor behavior in *Sgce* knockdown mouse model of DYT11.

| | |
|---|---|
| 0 | Not present – Animal does not spin |
| 1 | Mild – Animal spins intermittently or clearly struggles more than the wild-type |
| 2 | Moderate – Animal spins moderately, making several successive complete rotations |
| 3 | Severe – Animal spins vigorously for most of the duration of the observation |

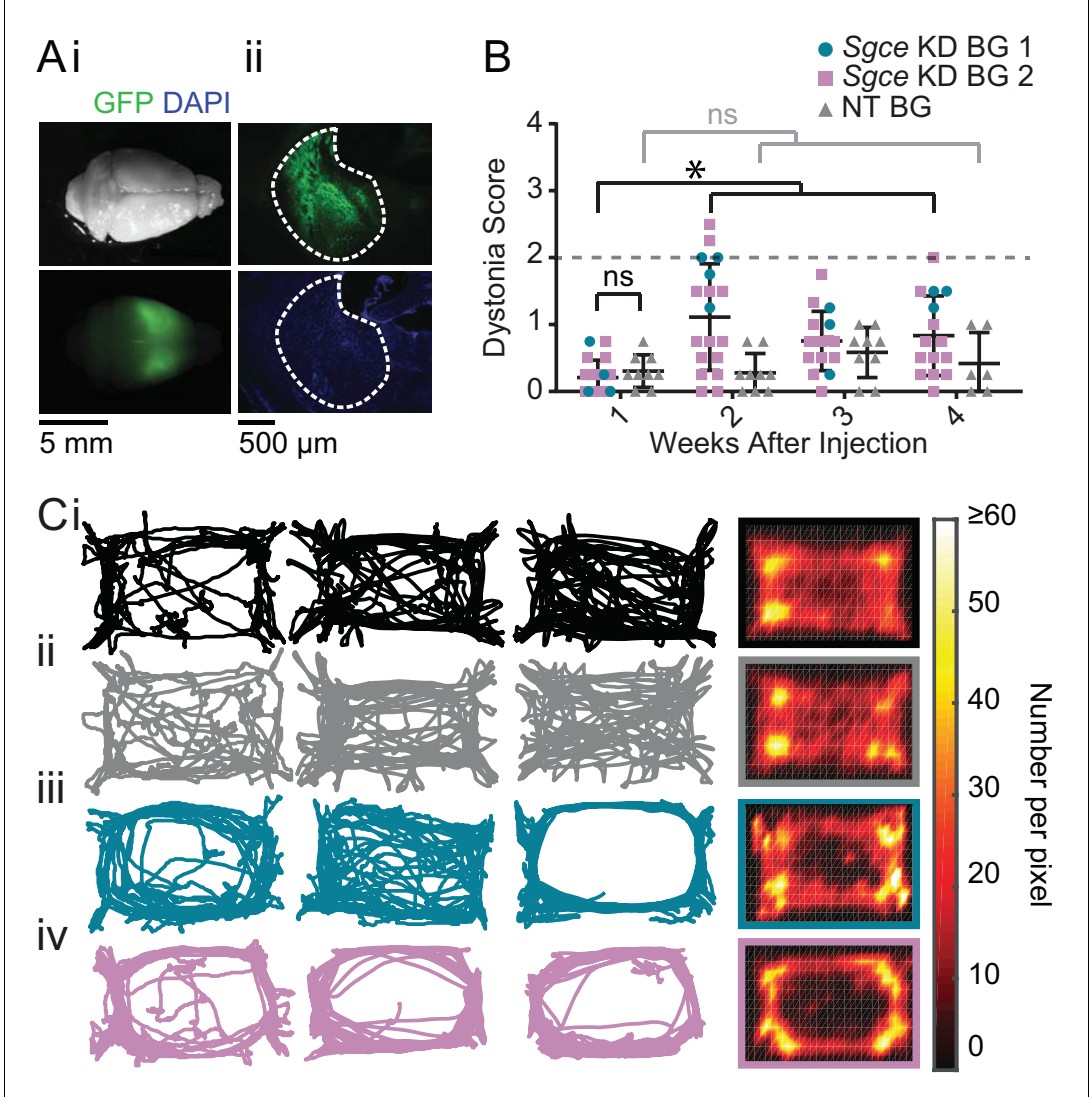

**Figure 2.** shRNA-mediated knockdown of *Sgce* in the basal ganglia causes motor abnormalities but does not cause overt dystonia. (**A**) Images of the whole brain (**i**) and coronal section (**ii**) from an *Sgce* KD BG mouse. (**B**) Injection of *Sgce* KD- or NT-shRNA into the basal ganglia did not produce dystonia, as indicated by a score greater than two on the Dystonia scale. (*Sgce* KD BG 1: N = 4; *Sgce* KD BG 2: N = 14; NT BG: N = 9). The dystonia scores for *Sgce* KD BG mice for time points of 2 weeks or more after injection were significantly different from the dystonia scores of the same animals at 1 week (Wilcoxon matched-pairs signed rank test, p<0.001). The dystonia scores of *Sgce* KD BG mice at 1 week after injection were not significantly different from NT BG mice at the same time point (t-test, Holm-Sidak method, p=0.36). (**C**) *Sgce* KD BG 1 (**iii**) and *Sgce* KD BG 2 mice (**iv**) appeared to ambulate more in the periphery of the open field chamber than wild-type (**i**) and NT BG (**ii**) mice. The first three columns show example tracks from individual mice. The last column depicts the average, which reflects the number of times the center of mass was detected at a pixel in the arena, and excludes frames where the animal did not move. (WT: N = 12, NT BG: N = 12, *Sgce* KD BG 1: N = 4; *Sgce* KD BG 2: N = 13).
The online version of this article includes the following figure supplement(s) for figure 2:

**Figure supplement 1.** *Sgce* KD BG mice spend qualitatively less time in the center of the open field with no significant change in the ratio of distance traveled.

**Figure supplement 2.** *Sgce* KD CB mice did not show a preference for the periphery and avoidance of the center.

symptoms in *Sgce* KD CB mice, then the output of the cerebellum, the DCN neurons, must be affected.

To that end, we performed extracellular single unit recordings from DCN neurons in awake, head-restrained *Sgce* KD CB or NT CB mice. We found that shRNA knockdown of *Sgce* mRNA in the cerebellum caused aberrant activity of cerebellar output neurons (***Figure 4***). Specifically, we found that the firing rate of DCN neurons in *Sgce* KD CB mice was reduced (NT CB = 62.9 ± 24.8

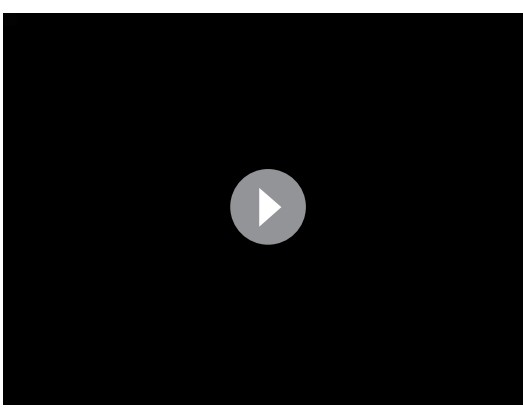

**Video 3.** Neither *Sgce* KD BG nor NT BG mice developed dystonia, as measured by a score of greater than or equal to two in the open field.
https://elifesciences.org/articles/52101#video3

spikes/s, n = 9, N = 4 and *Sgce* KD CB = 32.2 ± 19.5 spikes/s, n = 32, N = 8, Mean ± S.D.; Welch's t-test, p=0.0057). At the same time, the interspike interval coefficient of variation (ISICV) of DCN neurons was increased in *Sgce* KD CB mice (NT CB = 0.50 ± 0.16 and *Sgce* KD CB = 1.00 ± 0.61, Mean ± S.D.; Welch's t-test, p=0.0002). The ISICV, a frequently used measure of the regularity of the firing rate of a cell, is the standard deviation of the ISI divided by the mean ISI and thus dependent on the average firing rate of the cell. *Sgce* KD CB mice had a decreased average firing rate, which would result mathematically in a decreased ISICV if the standard deviation of the ISI was unchanged. The significant increase in the ISI CV in *Sgce* KD CB mice demonstrates that despite the decrease in average firing rate, the cells are firing with much greater variability in their ISIs. There were no significant differences in the mode firing rates of *Sgce* KD CB mice and NT CB mice (NT CB = 80.4 ± 36.6 and *Sgce* KD CB = 74.8 ± 53.5 spikes/s, Mean ± S. D.; Welch's t-test, p=0.73). The change in ISICV was primarily a consequence of more pauses in DCN neurons in *Sgce* KD CB mice, rather than a shift to high-frequency burst firing. There was a clear broadening of the curve of the ISI histogram in *Sgce* KD CB mice (*Figure 4F*), and no increase in the autocorrelation of neurons at time points very close to zero (*Figure 4G*).

The Purkinje cells, the principle cells of the cerebellar cortex and its sole output, provide powerful inhibitory synaptic input onto DCN neurons. The irregular firing activity observed in DCN neurons could partly be due to aberrant Purkinje cell input. To examine this possibility, we performed extracellular recordings from Purkinje cells in awake, head-restrained *Sgce* KD CB and NT CB mice (*Figure 5*). Similar to what was observed in DCN neurons, *Sgce* KD in the cerebellum reduced the average firing rate of Purkinje cells (NT CB = 53.3 ± 24.1 spikes/s, n = 30, N = 4 and *Sgce* KD CB = 39.2 ± 26.6 spikes/s, n = 57, N = 11, Mean ± S.D.; Welch's t-test, p=0.0028), and increased their ISICV (NT CB = 0.60 ± 0.25 and *Sgce* KD CB = 1.06 ± 0.57, Mean ± S.D.; Welch's t-test, p<0.0001). In contrast to DCN neurons, Purkinje cells in *Sgce* KD CB mice had an increased mode firing rate (NT CB = 77.5 ± 41.6 spikes/s and *Sgce* KD CB = 115.0 ± 99.4 spikes/s, Mean ± S. D.; Welch's t-test, p=0.0158). This is consistent in the rightward shift of the curve in the ISI histogram (*Figure 5F*) and the observation of longer pauses in DCN neurons. Like DCN neurons, Purkinje cells did not show an increase in the autocorrelation of neurons at time points very close to zero (*Figure 5G*), suggesting that the neurons are not necessarily burst firing at high frequencies, despite the increase in the predominant firing rate.

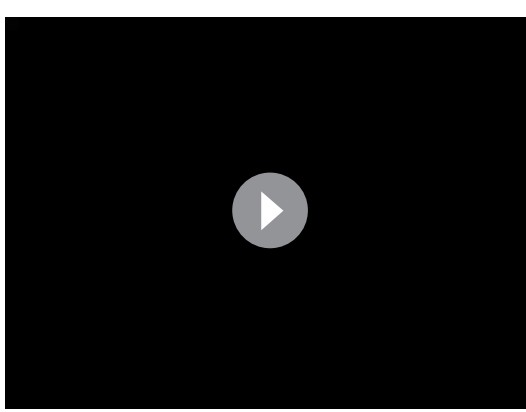

**Video 4.** EtOH relieves motor symptoms of *Sgce* KD CB mice, as measured on the Disability Scale (p<0.0001, 1way ANOVA, N = 16), Spinning Scale Scale (, p<0.0001, 1way ANOVA, N = 19), and Dystonia Scale (p<0.0001, 1way ANOVA, N = 16), but it does not relieve dystonic symptoms of mice injected with shRNA against *Tor1a*, an acute shRNA knockdown model of DYT1 (p=0.2391, 1way ANOVA, N = 5).
https://elifesciences.org/articles/52101#video4

## Discussion

The primary goal of this study was to test the hypothesis that acute knockdown of *Sgce*, the mouse homolog of the gene responsible for DYT11 in humans, in adult mice would more accurately model DYT11 by circumventing

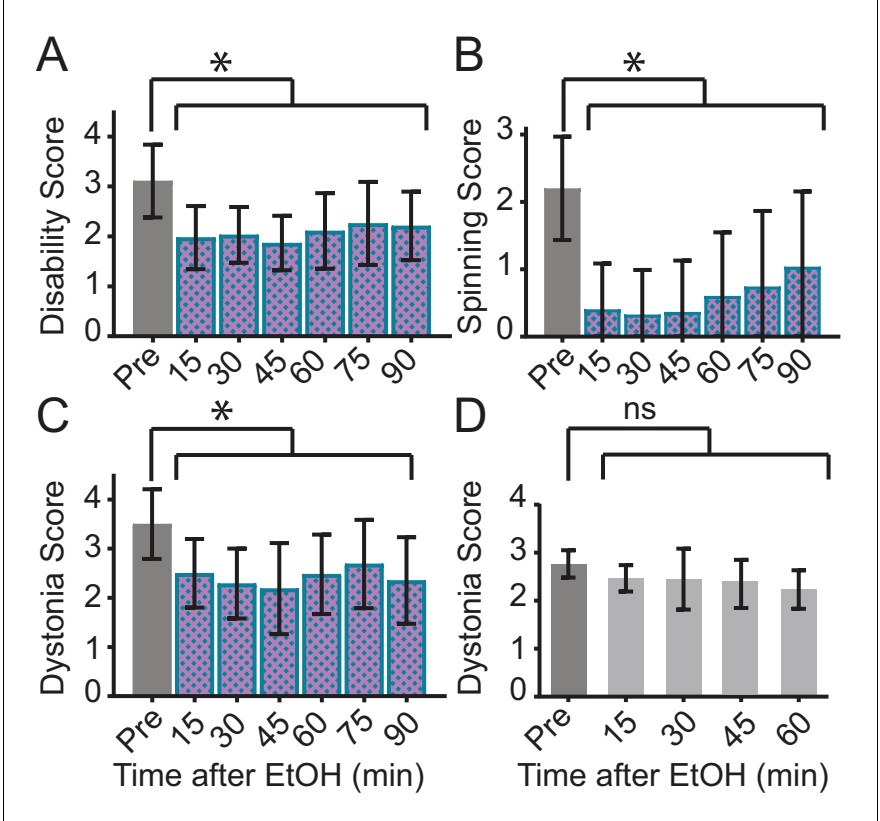

**Figure 3.** Ethanol relieves motor symptoms in *Sgce* KD CB, but not *Tor1a* KD CB, mice. (**A**) Disability score of *Sgce* KD CB mice after ethanol injection. Ethanol reduces the disability score of mice injected with shRNA against *Sgce*. Alleviation of symptoms persisted for up to 90 min after ethanol (p<0.0001, 1way ANOVA, Mean + S.D., N = 16). (**B**) Spinning score of *Sgce* KD CB mice after ethanol injection. Ethanol reduces the spinning score of mice injected with shRNA against *Sgce*. (p<0.0001, 1way ANOVA, Mean + S.D., N = 19). (**C**) Dystonia score of *Sgce* KD CB mice after ethanol injection. Ethanol significantly reduced the dystonia score of *Sgce* KD CB mice (p<0.0001, 1way ANOVA, Mean + S.D., N = 16). (**D**) Dystonia score of *Tor1a* KD mice after ethanol injection. Ethanol had no effect of the dystonia score, which reflects the primary symptoms caused by *Tor1a* knockdown, in mice injected with shRNA against *Tor1a* (p=0.2391, 1way ANOVA, Mean + S.D., N = 5).

The online version of this article includes the following figure supplement(s) for figure 3:

**Figure supplement 1.** The effect of EtOH on distance travelled varies among individual mice.

**Figure supplement 2.** Saline does not improve symptoms in *Sgce* KD CB mice.

compensation that might occur in transgenic mouse models where the protein is absent throughout the development of the brain. We found that acute, shRNA-mediated knockdown of *Sgce* in the cerebellum, but not the basal ganglia, was sufficient to induce dystonia and other motor symptoms in adult mice. Consistent with what has been observed in patients, the motor symptoms of *Sgce* KD CB mice were sensitive to alcohol, distinguishing this model from previous models of DYT11.

Previous genetic mouse models of DYT11 include both ubiquitous and cell-type specific *Sgce* knockout mice. Because the maternal *SGCE* allele is epigenetically silenced in humans, DYT11 is paternally inherited in the vast majority of cases. In mice, there is no expression of maternally-inherited *Sgce* (*Yokoi et al., 2005*). To reflect the inheritance pattern observed in humans, mouse models of DYT11 were generated by crossing male heterozygous *Sgce* knockout mice with wild-type females. Ubiquitous genetic knockout of *Sgce* resulted in subtle motor symptoms, including full body jerks and deficits in motor learning in a balance beam task (*Yokoi et al., 2006*). In contrast, *Sgce* KD CB mice, which in most symptomatic cases have less than half of wild-type *Sgce* levels, exhibit jerking movements and dystonic-like motor symptoms that are responsive to ethanol. Our success in generating symptomatic mice using an acute knockdown approach points to possible

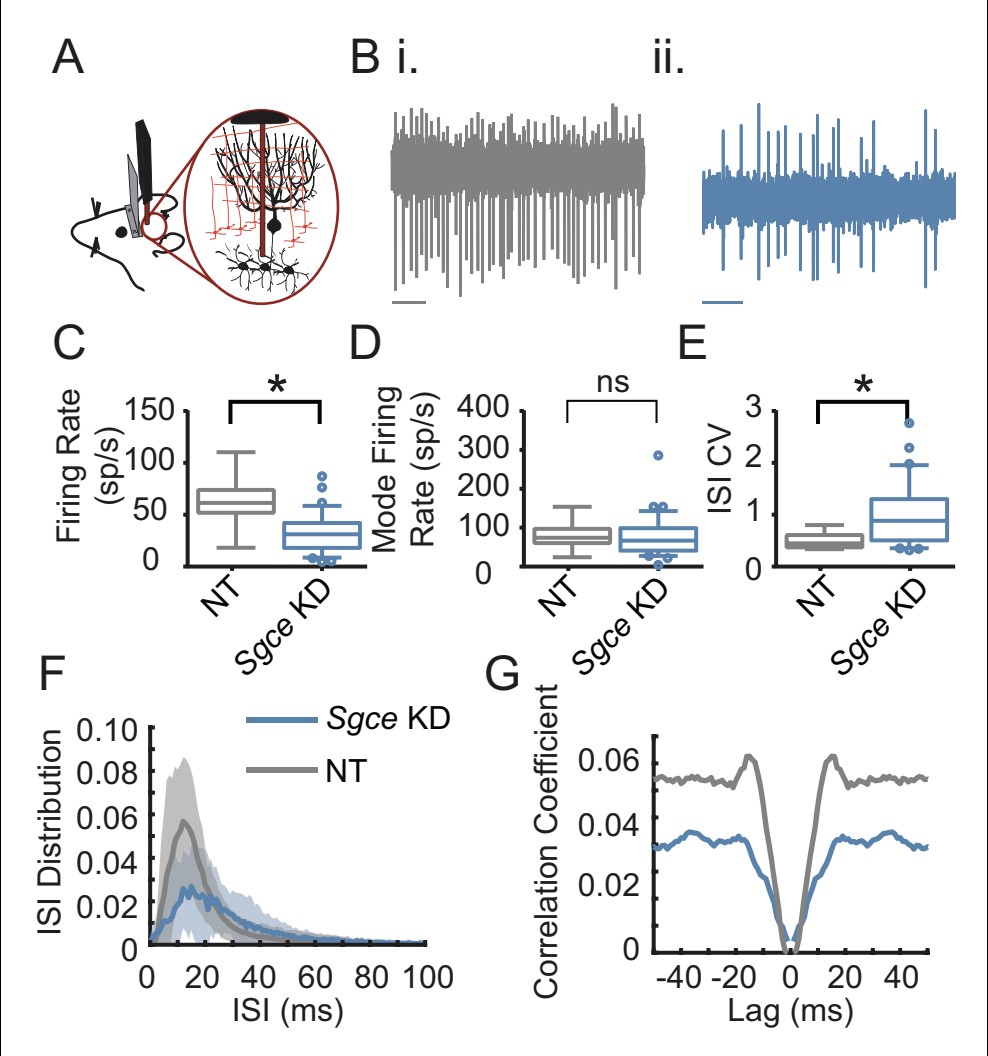

**Figure 4.** Cerebellar nuclei neurons fire aberrantly in *Sgce* KD CB mice. (**A**) Experimental schematic. Extracellular electrophysiological recordings were made from neurons in the cerebellar nuclei in awake, head-restrained mice. (**B**) Example traces from a mouse injected with non-targeting shRNA (**i**) and shRNA against *Sgce* (**ii**). Scale bar represents 100 ms. (**C**) Average firing rates of DCN neurons in NT CB and *Sgce* KD CB animals (NT CB = 62.9 ± 24.8 spikes/s, n = 9, N = 4 and *Sgce* KD CB = 32.2 ± 19.5 spikes/s, n = 32, N = 8, Mean ± S.D.; Welch's t-test, p=0.0057). (**D**) Mode firing rates of DCN neurons in NT CB and *Sgce* KD CB animals (NT CB = 80.4 ± 36.6 and *Sgce* KD CB = 74.8 ± 53.5 spikes/s, Mean ± S.D.; Welch's t-test, p=0.73). (**E**) Interspike interval coefficients of variation of DCN neurons in NT CB and *Sgce* KD CB animals (NT CB = 0.50 ± 0.16 and *Sgce* KD CB = 1.00 ± 0.61, Mean ± S.D.; Welch's t-test, p=0.0002). (**F**) Normalized ISI histogram of DCN neurons in NT CB and *Sgce* KD CB mice. (**G**) Autocorrelogram of DCN neurons in NT CB and *Sgce* KD CB mice.

rodent-specific, developmental compensation for loss of *Sgce* in previous genetic mouse models of DYT11 (*Xiao et al., 2017*; *Yokoi et al., 2006*; *Yokoi et al., 2012a*; *Yokoi et al., 2012b*). Indeed, a recently-developed genetic model of DYT11 that used gene-trap technology to knock down the main and brain-specific isoforms of *Sgce* observed dystonic-like posturing between postnatal days 14 and 16, an important period of Purkinje cell development (*Xiao et al., 2017*). These symptoms did not persist into adulthood, however, supporting the hypothesis that developmental compensation may occur in genetic knockout animals. One possible source of compensation is the upregulation of genes with related functions. It has been shown that ε-SG forms complexes with the other members of the sarcoglycan family (α, β, γ,δ, and ζ), and also with members of the dystrophin-associated glycoprotein complex (DGC) in the brain (*Waite et al., 2016*). It is possible that upregulation

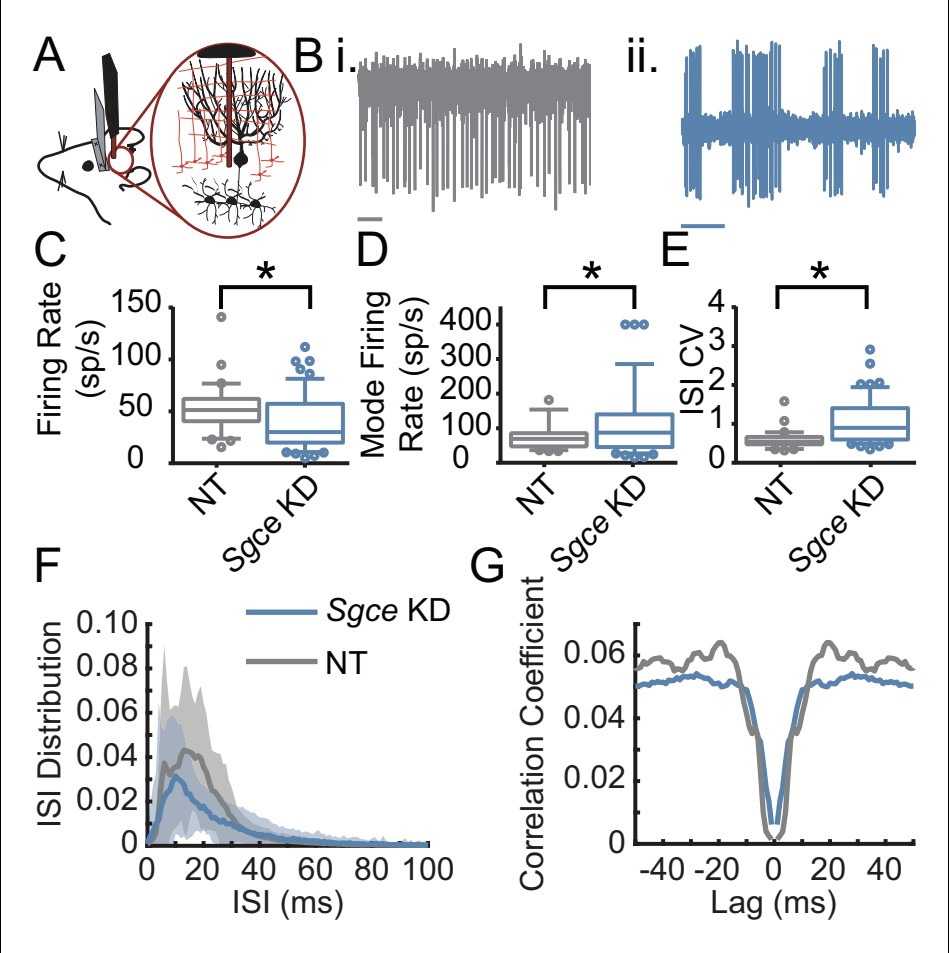

**Figure 5.** Purkinje cells fire aberrantly in *Sgce* KD CB mice. (**A**) Experimental schematic. Extracellular electrophysiological recordings were made from Purkinje cells in awake, head-restrained mice. (**B**) Example traces of Purkinje cells from an NT CB (**i**) or *Sgce* KD CB (**ii**) mouse. Scale bar represents 100 ms. (**C**) Average firing rates of Purkinje cells in NT CB and *Sgce* KD CB animals (NT CB = 53.3 ± 24.1 spikes/s, n = 30, N = 4 and *Sgce* KD CB = 39.2 ± 26.6 spikes/s, n = 57, N = 11, Mean ± S.D.; Welch's t-test, p=0.0028). (**D**) Mode firing rates of Purkinje cells in NT CB and *Sgce* KD CB animals (NT CB = 77.5 ± 41.6 spikes/s and *Sgce* KD CB = 115.0 ± 99.4 spikes/s, Mean ± S.D.; Welch's t-test, p=0.0158). (**E**) Interspike interval coefficients of variation of Purkinje cells in NT CB and *Sgce* KD CB animals (NT CB = 0.60 ± 0.25 and *Sgce* KD CB = 1.06 ± 0.57, Mean ± S.D.; Welch's t-test, p<0.0001). (**F**) Normalized ISI histogram of Purkinje cells in NT CB and *Sgce* KD CB mice. (**G**) Autocorrelogram of Purkinje cells in NT CB and *Sgce* KD CB mice.

of the other members in this complex during development could compensate for loss of *Sgce* in mice. The shRNA-mediated knockdown approach circumvents possible developmental compensation by knocking down the protein in the adult mouse and may be more effective than previous models of DYT11 for understanding how loss of *Sgce* leads to motor symptoms.

The model presented here is the first description of a mouse model of DYT11 that recapitulates the salient features of DYT11, namely, jerking, myoclonic-like movements and dystonia that were responsive to alcohol. However, in addition to overt dystonia, *Sgce* KD CB mice also had difficulty ambulating normally and showed a mildly ataxic gait. While ataxia is uncommon in DYT11 patients, it has been observed in individuals with SGCE mutations (*Drivenes et al., 2015*; *Sun et al., 2016*). It is also worth noting that the most obvious motor symptoms in *Sgce* KD CB mice were observed in the hind limbs and tails of the animals. While lower limb involvement has been reported in some DYT11 cases (*Kobylecki et al., 2014*), involvement of the trunk and upper extremities is far more common (*Asmus et al., 2002*). It is possible that our method of evaluating symptom severity based on activity in the open field biased us towards highlighting motor dysfunction in the hind limbs and

the tail, and limited our ability to detect more severe motor symptoms in the forelimbs and trunk of *Sgce* KD CB mice that can be more carefully scrutinized in a skilled forelimb reaching task. Lastly, *Sgce* KD CB mice exhibited spinning when suspended by the tail. While there is no corresponding human symptom, understanding how loss of *Sgce* contributes to this symptom and how alcohol improves it may yield important insights into *Sgce* function.

The acute knockdown strategy further enables us to examine which parts of the brain are responsible for myoclonus and dystonia in DYT11. Our findings implicate the cerebellum as a central structure contributing to the motor symptoms in this disorder, in agreement with findings in human patients (*Beukers et al., 2010*; *Carbon et al., 2013*; *Nitschke et al., 2006*; *van der Meer et al., 2012*; *van der Salm et al., 2013*) and consistent with what has been reported in multiple dystonic rodent models (*Campbell and Hess, 1998*; *Campbell et al., 1999*; *LeDoux et al., 1993*; *LeDoux et al., 1995*; *Neychev et al., 2008*). We found that knockdown of *Sgce* specifically in the cerebellum led to dystonic-like movements that were responsive to alcohol. This suggests that the motor symptoms are a consequence of cerebellar dysfunction, supported by the aberrant activity recorded from DCN neurons and Purkinje cells, and that alcohol may act through the cerebellum to ameliorate them. We found that both Purkinje cells and DCN neurons exhibited irregular firing rates in vivo, indicated by an increased ISICV. These findings are consistent with our observations in many animal models of movement disorders, including ataxia and dystonia. In general, the more symptomatic the animal, the more irregular the cerebellar activity. In these animals, we found that irregular activity of Purkinje cells and DCN neurons was correlated with the presence of dystonia (*Fremont et al., 2014*; *Fremont et al., 2017*), in agreement with what has been reported in other rodent models of dystonia (*Fremont et al., 2014*; *Fremont et al., 2015*; *Isaksen et al., 2017*; *LeDoux et al., 1998*; *LeDoux and Lorden, 1998*; *LeDoux and Lorden, 2002*; *White and Sillitoe, 2017*). While *Sgce* KD CB mice exhibit dystonia, we did not perform EMG recordings in agonist and antagonist muscles at the time of recording. We were thus unable to determine whether highly irregular firing was time-locked to dystonic episodes. The increased ISI CV in *Sgce* KD CB mice thus reflects a combination of acute dystonic, and persistent abnormal motor activity in these animals.

Cerebellar contribution to DYT11 was examined previously in a Purkinje cell-specific knockout of *Sgce*. In contrast to our findings and what was reported in ubiquitous *Sgce* null mice, the symptoms of Purkinje cell knockout mice were quite mild; they exhibited a small deficit in motor learning but had no robust motor abnormalities or jerking movements, suggesting brain regions other than the cerebellum might contribute to these motor symptoms (*Yokoi et al., 2012a*). However, one major difference between the present work and previous work is that our knockdown is not specific to Purkinje cells. It is possible that *Sgce*, which is expressed in DCN neurons in addition to Purkinje cells (*Ritz et al., 2011*), is important for the function of cerebellar output neurons. This is consistent with our finding that the average firing rate is decreased and the ISICV is increased in DCN neurons in *Sgce* KD CB mice. The effect of *Sgce* knockdown in DCN neurons may provide an alternative explanation for the more severe motor phenotype of *Sgce* null mice compared to the Purkinje cell-specific knockout. Further experiments are required to determine whether it is the intrinsic activity of DCN neurons that contributes to motor symptoms, or whether it is their synaptic inputs, including Purkinje cells, which were also affected in our study, that lead to erratic cerebellar output and subsequent motor abnormalities. The symptomatic mouse model produced by acute knockdown of *Sgce* provides a unique opportunity to address these questions.

Because so much is known about the microcircuitry of the cerebellum, as well as the channels that contribute to the intrinsic firing rate of cerebellar neurons, understanding how the electrical properties of cerebellar neurons change in *Sgce* KD CB mice will provide insight into the function of ε-SG under normal conditions. Understanding how alcohol acts to relieve dystonic symptoms would identify therapeutic targets for DYT11 in the cerebellum and help elucidate the mechanisms by which alcohol influences cerebellar output. A number of potential targets of ethanol in the brain have been described (*Harris et al., 2008*; *Narahashi et al., 2001*). One well-described and intensely-debated target of ethanol is the delta subunit-containing extrasynaptic $GABA_A$ receptor (*Hanchar et al., 2005*; *Wallner et al., 2003*), which is responsible for maintaining tonic inhibitory current in cerebellar granule cells (*Stell et al., 2003*). Modulation of inhibitory currents by ethanol via extrasynaptic $GABA_A$ receptors is a potential mechanism for restoring the firing rate and regularity of cerebellar neurons, thereby relieving dystonia.

While our findings are consistent with a growing body of evidence in support of a role for the cerebellum in DYT11, our results do not rule out the possibility that other brain regions also contribute to the symptoms observed in these mice. Knockdown of *Sgce* in the basal ganglia induced subtle motor defects in the mice. Frequently, *Sgce* KD BG mice exhibited increased kinetic behavior specifically in the periphery of the open field chamber. An intriguing possibility is that this behavior is related more to the non-motor symptoms associated with DYT11, including obsessive compulsive disorder and anxiety.

It is now well-established that there are a number of direct connections between the cerebellum and the basal ganglia. Given the extensive connections between these brain regions and the contributions of both structures to motor control, it has been suggested that movement disorders should be considered less as disorders of a particular brain region, and more as disorders of the motor circuit. This is because dysfunction in one node, that is the cerebellum or basal ganglia, can influence the activity of other nodes (*Bostan and Strick, 2018*). With that in mind, two things become very clear. First, it is important from a basic science perspective to understand the site of initial dysfunction in the different dystonias in order to truly understand how abnormal movement results from a genetic mutation. In the case of DYT11, our approach suggests that loss-of-function of *Sgce* can lead to irregular activity of the cerebellum, resulting in motor symptoms. Understanding this link between genetic mutation and motor symptoms will yield important information about cerebellar function in health and disease. Second, the result of the interconnectivity of different brain structures is that there are multiple sites for therapeutic intervention. It has been shown that deep brain stimulation of the GPi, as well as the ventral intermediate nucleus of the thalamus (VIM) can improve symptoms in most DYT11 patients, with or without *SGCE* mutations (*Azoulay-Zyss et al., 2011*; *Fernández-Pajarín et al., 2016*; *Gruber et al., 2010*; *Kosutzka et al., 2019*; *Rocha et al., 2016*; *Rughani and Lozano, 2013*). It is possible that DBS of the GPi or VIM may influence cerebellar-recipient regions of the thalamus, disrupting the irregular output from the cerebellum and, in this way, improving motor symptoms. Indeed, there is considerable overlap of cerebellar and pallidal projections in the thalamus (*Hintzen et al., 2018*). Another interesting possibility, consistent with pervious work on the anatomy of cerebellar-basal ganglia connectivity, is that irregular activity of the cerebellum subsequently drives irregular activity in the striatum, leading to motor symptoms in dystonia. DBS of the GPi may relieve dystonic symptoms by disrupting the abnormal output from the basal ganglia, caused in part by cerebellar dysfunction, and thereby restoring some functionality to the system.

Taken together, our studies show that loss of function of *Sgce* in the cerebellum results in aberrant cerebellar output which in turn causes motor dysfunction, including jerking movements and dystonia, suggesting that the cerebellum may be a major of site of dysfunction in DYT11. The efficacy of ethanol in reducing the severity of the motor symptoms in *Sgce* KD CB mice, as it does in DYT11 patients, further validates the model. Thus, the acute shRNA-mediated knockdown model of DYT11 described here not only provides a platform for further scrutiny of the mechanisms by which loss of ε-SG causes abnormal neuronal activity and motor dysfunction in DYT11, but also provides an opportunity to study how alcohol exerts its beneficial effects and, subsequently, identify alternative therapeutic strategies that mimic the effects of alcohol without its addictive and neurodegenerative consequences.

## Materials and methods

**Key resources table**

| Reagent type (species) or resource | Designation | Source or reference | Identifier(s) | Additional information |
|---|---|---|---|---|
| Sequenced-based reagent | *Sgce* KD 1 | RNAi Consortium (sequence) Virovek (virus) | TRCN0000119308; Lot# 12–183 | shRNA; 5'-CCGGGCCGAGACT ATTACACGGATTCTCGAGAATCCGT GTAATAGTCTCGGCTTTTTG-3' |
| Sequenced-based reagent | *Sgce* KD 2 | RNAi Consortium (sequence) Virovek (virus) | TRCN0000119307; Lot# 13–009 | shRNA; 5'-CCGGCCCACTGTG TTGAGAACCAAACTCGAGTTTGGT TCTCAACACAGTGGGTTTTTG-3' |

*Continued on next page*

*Continued*

| Reagent type (species) or resource | Designation | Source or reference | Identifier(s) | Additional information |
|---|---|---|---|---|
| Sequenced-based reagent | NT | Virovek | Lot# 13–234 | shRNA; 5'-GAGGATCAAATTG ATAGTAAACCGTTTTGGCCACTGACT GACGGTTTACTATCAATTTGATCCTCTTTTT-3' |
| Sequenced-based reagent | NT | Virovek | Lot# 13–037 | shRNA; 5'-CCAACTACCCGAA CTATTATTCAAGAGATAATAGTTC GGGTAGTTGGCATTTTTT-3' |
| Commercial assay or kit | iTaq universal SYBR Green reaction mix | BioRad | Catalog #172–5151 | |
| Commercial assay or kit | iScript reverse transcriptase | BioRad | Catalog #172–5151 | |
| Sequenced-based reagent | *Sgce* Forward Primer | PrimerBank | PrimerBank ID: 31981494a1 | CGGATTCTTTGAAAAGCCGAGA |
| Sequenced-based reagent | *Sgce* Reverse Primer | PrimerBank | PrimerBank ID: 31981494a1 | GTCTGTGTGCATGGGAGGTAT |
| Sequenced-based reagent | *Gapdh* Forward Primer | PrimerBank | PrimerBank ID: 6679937a1 | AGGTCGGTGTGAACGGATTTG |
| Sequenced-based reagent | *Gapdh* Reverse Primer | PrimerBank | PrimerBank ID: 6679937a1 | TGTAGACCATGTAGTTGAGGTCA |
| Antibody | Alexa 488, goat anti rabbit secondary | Invitrogen Life Technologies; Thermo Fisher Scientific | Cat# A-11008, RRID:AB_143165 | IF 1:400 |
| Antibody | GFP Tag (rabbit polyclonal) | Invitrogen Molecular Probes; Thermo Fisher Scientific | Cat# A-11122, RRID:AB_221569 | IF 1:250 |
| Other | DAPI stain | Invitrogen Life Technologies | H1399 | 1:2000 |
| Software, algorithm | LabVIEW | National Instruments | RRID: SCR_014325 | |
| Software, algorithm | GraphPad Prism 7 | GraphPad Software | RRID: SCR_002798 | |
| Other | OptiBond All-In-One | Kerr | 33381 | Adhesive |
| Other | Charisma | Heraeus Kulzer | NA | Adhesive |
| Other | Metabond | Parkell | S380 | C and B Metabond Quick Adhesive Cement System |

## Method details

Experiments were performed male and female C57BL/6 mice at least 6 weeks old in accordance with the guidelines set by Albert Einstein College of Medicine.

## shRNA sequences

Two different shRNAs against unique parts of the *Sgce* sequence generated originally by the RNAi consortium were identified and used in these experiments (Key Resources Table). AAV9 compatible plasmids and AAV9 virus containing each shRNA were generated commercially (Virovek, Hayward, CA, AAV9-U6-shRNA119308-CMV-GFP (Lot# 12–183); AAV9-U6-shRNA_SGCE-CMV-GFP (Lot# 13–009), respectively). Each virus has an average titer of $2 \times 10^{13}$ vg/ml. A 0.22 µm filter sterilized solution containing the virus in DPBS buffer with 0.001% pluronic F-68 was directly injected into the cerebellum or basal ganglia of adult mice. Control AAV9 containing non-targeted (NT) shRNAs under the same promoter containing the same CMV driven GFP at equivalent titer were also purchased (Key Resources Table).

## Stereotaxic injection of shRNA

Male and female C57BL/6 mice at least 6 weeks of age were anesthetized with 4% isoflurane and placed on a stereotaxic frame (David Kopf Instruments, Tujunga, CA). Isoflurane anesthesia was maintained at 1.5–2% throughout the surgery, which was sufficient to prevent the animal from responding to a toe or tail pinch. A midline incision was made and the surface of the skull was cleaned of all connective tissue. Craniotomies were made over the cerebellum or basal ganglia, and 2 µl shRNA (AAV-SGCEshRNA-GFP or AAV-shNT-GFP) was injected into each injection site at a rate of 0.15 µl/min. After each injection, the syringe was retracted 50 µm and left for at least 5 min. For the cerebellum, the following coordinates were used: −6.00 AP, 0 ML, −1.5 DV and −6.96 AP, 0 ML, and 1.5 DV (cerebellar cortex) and −6.00 AP, ±1.8 ML, −2.3 DV (cerebellar nuclei). For the basal ganglia, the following coordinates were used: +0.5 AP, ±2.0 ML, −2.5 DV (striatum) and −0.5 AP, ±2.5 ML, −3.5 DV (GPi). A total of 8 µl was injected into the brain. In these experiments, every effort was made to include all injected animals. However, on rare occasions, an animal was excluded if post-hoc histology revealed a lack of expression of the virus in the targeted brain area or expression of the virus outside the targeted brain area.

## qRT-PCR

Mice were anesthetized with isoflurane until breathing slowed to 1 breath per 3 s and the animal was completely unresponsive to tail- or toe pinch. The animal was then quickly decapitated. The cerebellum was rapidly dissected, placed in a 1.5 mL Eppindorf tube, and frozen in liquid nitrogen. Samples were then transported on dry ice to the Molecular Cytogenetic Core at Albert Einstein College of Medicine, where RNA extraction was performed. qRT-PCR was carried out on the CFX96 Touch Real-Time PCR Detection System (BioRad) with the iTaq Universal SYBR Green One-Step Kit (BioRad, Catalog# 172–5151). In total, RNA from the cerebella of 6 wild-type, 5 NT, 13 *Sgce*KD1, and 7 *Sgce*KD2 mice were examined. Primer sequences for *Sgce* and *Gapdh* were identified through PrimerBank (*Spandidos et al., 2008*; *Spandidos et al., 2010*; *Wang and Seed, 2003*), and oligonucleotides were commercially generated (Eurofins MWG Operon Oligos Tool, Thermo Fisher Scientific). Each 10 µl reaction contained 300 nM each of forward and reverse primer and 100 ng RNA. All reactions were performed in duplicate or triplicate.

Analysis was performed using the $2^{(-\Delta\Delta C_T)}$ method (*Livak and Schmittgen, 2001*). In order to be used for analysis, the threshold cycle ($C_T$) of two replicates had to be within 0.4 cycles of each other (standard deviation $\leq 0.283$). The median $C_T$ for each group of replicates was used. $\Delta C_T$ was calculated by subtracting the $C_T$ for *Gapdh*, a housekeeping gene control, from the $C_T$ for *Sgce* for each RNA sample. $\Delta\Delta C_T$ was calculated by subtracting the mean $C_T$ for all WT samples from the $C_T$ for each sample. Relative RNA quantity was then determined by calculating $2^{(-\Delta\Delta C_T)}$.

## Immunohistochemistry

Immunohistochemistry of GFP was used to confirm expression of the virus, which encodes GFP, in the targeted brain region. On rare occasions, animals were excluded when there was either no clear GFP expression in the target area or when there was GFP expressed in regions outside of the target area. Mice were anesthetized with isoflurane and transcardially perfused with phosphate buffered saline (PBS, Thermo Fisher Scientific, Waltham, MA) followed by 4% paraformaldehyde (PFA, Acros Organics, Thermo Fisher Scientific). The brains were dissected and fixed overnight in 4% PFA at 4°C. They were then rehydrated with 30% sucrose for 24–48 hr, rapidly frozen in Optimal Cutting Temperature Compound (OCT, Tissue-Tek) on dry ice, and stored at −80°C. Brains were cut on a cryostat (Leica CM3050 S) into 30 µm sections. Sections were stained with primary antibody against GFP (rabbit, 1:250, Molecular probes, A11122), followed by secondary antibody Alexa 488 (1:400, goat anti rabbit, Invitrogen Life Technologies A11008). Nuclei were labeled with DAPI (1:2000, Hoescht 33342, Invitrogen Life Technologies H1399). Images of sections were captured under a standard fluorescent microscope (Zeiss Axioskop 2 Plus).

## Dystonia scale

The presence and severity of dystonia was quantified using a previously published scale (*Calderon et al., 2011*). Briefly, 0 = normal behavior; 1 = abnormal motor behavior, no dystonic postures; 2 = mild motor impairment, dystonic-like postures when disturbed; 3 = moderate impairment,

frequent spontaneous dystonic postures; 4 = severe impairment, sustained dystonic postures. The videos were assessed independently by four observers who were blinded to the animal's condition. The observers were trained with a video set containing representative examples for each score in which key characteristics were highlighted.

### Disability scale

In order to more accurately examine the range of motor symptoms observed in *Sgce* KD CB mice and measure the response of motor symptoms to ethanol, we generated the Disability Scale, which considers the frequency of both sustained dystonic-like postures and repetitive movements, and the extent of motor impairment. Behavior in the open field was scored by four colleagues blind to the condition of the animal as follows: 0 = Normal, the animal has no observable motor deficit; 1 = slight motor disability, the animal may exhibit unsteady gait or uncoordinated movement, but does not have any repetitive movements or sustained postures; 2 = Mild motor disability, ambulation is mildly impaired, the animal exhibits wide stance, very unsteady gait, back-walking, and/or occasional repetitive movements or sustained postures; 3 = Moderate motor disability, ambulation is moderately impaired, to the extent that the animal does not properly ambulate, drags itself on its side, and may exhibit frequent sustained dystonic-like postures or repetitive movements; and 4 = severe motor disability, the animal is unable to properly ambulate for most of the duration of the observation, exhibits frequent repetitive movements, and/or sustained dystonic-like postures.

### Spinning scale

*Sgce* KD CB mice exhibited abnormal spinning when suspended by the tail. To quantify this spinning behavior, short video clips of the animal suspended by the tail were shown to four observers blind to the condition of the animal and scored as follows: 0 = Not present, the animal does not spin; 1 = mild, the animal spins but not rapidly and only for parts of the video; 2 = moderate, the animal spins rapidly, but not for the full duration of the video; and 3 = severe, the animal is spinning rapidly for the full duration of the video (*Table 2*, *Video 5*).

### Quantitative open field analysis

Videos of equal length were analyzed using EthoVision XT 14, a commercially available tracking software program (Noldus).

### Ethanol and saline injections

A working solution of 0.2 g/mL ethanol (EtOH) in saline was prepared. After a 5 min baseline period in the open field, *Sgce* KD CB animals were subcutaneously injected with ethanol at a volume of 200 µl for a 20 g mouse, for a final dose of 2 g/kg EtOH. During control trials, animals were injected with an equal volume of physiological saline. The behavior in the open field was recorded every 15 min for 90 min following EtOH injection. *Tor1a* KD CB mice were generated exactly as previously described (*Fremont et al., 2017*). These animals were given the same ethanol treatment, and their behavior in the open field was recorded every 15 min for 60 min following EtOH injection. Saline-injected animals were followed up to 60 min following the injection.

### In vivo electrophysiology

An L-shaped or flat titanium bracket was affixed to the skull of *Sgce* KD CB or NT CB mice with Charisma (Heraeus Kulzer) or Metabond (C and B Metabond Quick Adhesive Cement System, Parkell, S380) and dental cement. Craniotomies approximately 500 µm in diameter were made over the cerebellum for neuronal recordings. Recording sites were covered with a silicone

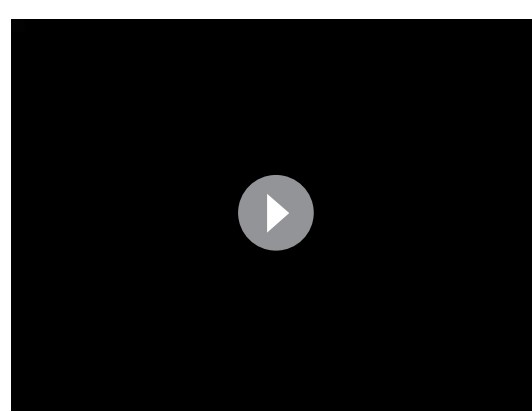

**Video 5.** Saline had no effect on the motor symptoms of *Sgce* KD CB mice (p=0.9517, 1way ANOVA, N = 6).
https://elifesciences.org/articles/52101#video5

adhesive (KWIK-SIL, WPI). Dental cement was used to construct a well around the recording site in order to hold saline during recording sessions.

Extracellular electrophysiological recordings were made from well-isolated single units using a tungsten electrode (Thomas Recording, 2–3 MΩ), which was advanced into the cerebellum until either the Purkinje cell layer or the DCN was reached. Purkinje cells were identified by location, firing rate, and the presence of complex spikes. DCN neurons were identified primarily by location and firing rate. Neurophysiological signals were amplified 2000x using a custom built amplifier and digitized at 20 kHz using a National Instruments BNC-2110. Waveforms were sorted offline by principal component analysis (Plexon).

## Statistics

GraphPad Prism 7 (GraphPad Software) was used to perform all statistical analyses. Data were assessed for normality using Shapiro-Wilk normality test. Non-normal data were compared using a non-parametric Wilcoxon test. Between group analysis was performed using the Holm-Sidak method. Each row was analyzed individually, without assuming a consistent SD. Normally distributed data sets were statistically with a two-tailed, paired Student's t-test or one-way ANOVA with Tukey's correction for multiple comparisons. Data are reported in text as mean ± S.D. unless otherwise stated.

## Acknowledgements

We would like to acknowledge the scorers of the open field videos for all behavioral experiments, as well as our funding source, the National Institute of Neurological Disorders and Stroke (NINDS) of the NIH. We would further like to acknowledge that the affinity-purified antibody against ε-SG was a kind gift from Dr. Kevin Campbell (HHMI, University of Iowa).

## Additional information

### Funding

| Funder | Grant reference number | Author |
| --- | --- | --- |
| National Institute of Neurological Disorders and Stroke | NS105470 | Kamran Khodakhah |
| National Institute of Neurological Disorders and Stroke | NS089716 | Samantha Washburn |

The funders had no role in study design, data collection and interpretation, or the decision to submit the work for publication.

### Author contributions

Samantha Washburn, Conceptualization, Data curation, Formal analysis, Funding acquisition, Investigation, Methodology, Project administration; Rachel Fremont, Conceptualization, Data curation, Formal analysis, Investigation, Methodology, Project administration; Maria Camila Moreno-Escobar, Chantal Angueyra, Formal analysis, Investigation; Kamran Khodakhah, Conceptualization, Resources, Supervision, Funding acquisition, Investigation, Methodology, Project administration

### Author ORCIDs

Samantha Washburn (ID) https://orcid.org/0000-0001-8317-1558
Kamran Khodakhah (ID) https://orcid.org/0000-0001-7905-5335

### Ethics

Animal experimentation: This study was performed in strict accordance with the recommendations in the Guide for the Care and Use of Laboratory Animals of the National Institutes of Health. The protocol was approved by the Committee on the Ethics of Animal Experiments of the Albert Einstein College of Medicine (Permit Number:20160805). All surgery was performed under isoflurane anesthesia, and every effort was made to minimize suffering.

## Decision letter and Author response

Decision letter https://doi.org/10.7554/eLife.52101.sa1
Author response https://doi.org/10.7554/eLife.52101.sa2

---

# Additional files

### Supplementary files
• Transparent reporting form

### Data availability
All data generated or analysed during this study are included in the manuscript and supporting files.

---

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
