## [Decision Letter]

Thank you for submitting your article "Acute cerebellar knockdown of *Sgce* reproduces salient features of Myoclonus-dystonia (DYT11) in mice" for consideration by *eLife*. Your article has been reviewed by three peer reviewers, and the evaluation has been overseen by Louis Ptacek as Reviewing Editor and Gary Westbrook as the Senior Editor. The following individuals involved in review of your submission have agreed to reveal their identity: Mark Ledoux (Reviewer #2). The reviewers have discussed the reviews with one another and the Reviewing Editor has drafted this decision to help you prepare a revised submission.

Summary:

The authors used an shRNA knockdown approach to generate a phenotypically penetrant mouse model of myoclonus-dystonia (alcohol-responsive dystonia), DYT11. Previous transgenic mouse models did not have dystonic postures typical in this condition. Moreover, the authors show that cerebellar knockdown of the gene produces dystonia, whereas basal ganglia knockdown results in more minor motor deficits. They go on to show that Purkinje cells and Deep Cerebellar Nucleus (DCN) neurons show abnormal rates and patterns of activity in the model. These results are exciting for two reasons. First, they provide a new, phenotypically penetrant mouse model of DYT11, which will permit future physiological, pharmacological and other studies to elucidate both the cellular mechanisms and test potential therapeutics for the disorder. Second, they corroborate this group's prior studies in other forms of dystonia, which suggest that the cerebellum is likely to be a primary site of cellular dysfunction in many forms of dystonia, and that abnormal patterns of Purkinje cell/DCN neuronal activity are a signature of this movement disorder.

Despite the excitement and potential impact of this study, there are some methodological issues that weaken their findings, and addressing these with some clarifications and minor additional experiments would make the paper a much more robust basis for future studies.

Essential revisions:

1) To strengthen the causal relationship between knockdown of epsilon sarcoglycan, it would be very helpful if the authors plotted individual mice in their cohort, comparing either SGCE protein levels or a surrogate marker, such as GFP expression associated with the shRNA. Some of the mice without symptoms may have had a more limited infection zone within the cerebellum, for example. It appears the individual protein levels are available, given the dots overlaying the Western blot results in Figure 1.

2) The authors mention that DYT11 is a dominant disorder – does that mean that humans with the disorder only carry one mutant gene? Do humans have about 50% of the protein expression, and haploinsufficiency produces disease symptoms? Or is it a toxic gain of function mutation? This issue merits some discussion in the text, and how their model may or may not replicate the human state in terms of protein.

3) The hyperactivity phenotype alluded to in the text (for the basal ganglia knockdown mice) is very interesting. Can the authors please include this data (it says "data not shown"). Quantitation of average velocity, rearing, or other hyperactivity phenotypes could be helpful. Video of each session could also be analyzed for time spent in the center versus periphery of the chamber, as this represents a phenotype linked to anxiety manipulations in mice, and could implicate the basal ganglia in the nonmotor aspects of the syndrome (as the authors suggest in the discussion).

4) The fact that ethanol relieves some of the motor symptoms in the cerebellar knockdown mice is compelling. However, it is not clear on why the dystonia score is noted in the original description of the mice (Figure 1), but the disability score and spinning score are noted in the ethanol experiments (Figure 3). Could the dystonia score also be plotted in Figure 3? I note that Dystonia score is displayed for the experiments using tor1A knockdown mice, so it would be best to include the same outcome measure for SGCE mice. Were saline injections done as controls for the effects of time and injection/handling itself on SGCE knockdown mice in the open field? The inclusion of open field distance or velocity with saline and ethanol would greatly strengthen the conclusions from these experiments.

5) The single-unit recordings from cerebellar neurons are interesting vis a vis the circuit mechanism of dystonia. Though the animals were head-fixed, were the authors able to note any dystonic posturing during the recordings? And if so, did the firing vary during dystonic episodes? This might help get at whether this is a fixed physiological abnormality and/or there are acute changes in cerebellar firing during dystonic movements. Likewise, did they note relationships between the severity of the motor phenotype in the open field and the degree of physiological abnormality noted during the single-unit recordings?

[Editors' note: further revisions were requested prior to acceptance, as described below.]

Thank you for resubmitting your work entitled "Acute cerebellar knockdown of *Sgce* reproduces salient features of Myoclonus-dystonia (DYT11) in mice" for further consideration at *eLife*. Your revised article has been favorably evaluated by Gary Westbrook (Senior Editor), a Reviewing Editor, and three reviewers. The manuscript has been improved but there are some small remaining issues to be addressed before acceptance, as outlined below:

Reviewer #1:

The authors did a good job of addressing the reviewer comments/requests and explained the limitations when appropriate. While they were not able to do everything requested, they made a good effort and these are not easy studies to perform. I appreciated the additional introduction and discussion which help to motivate their study and also fit their findings into the larger literature regarding this form of dystonia. I also appreciate the additional data they collected or have now included in the figures. I think this manuscript is now appropriate for publication.

Reviewer #2:

Overall, the revised manuscript is much improved. The authors have appropriately addressed my concerns and I have no additional queries.

Reviewer #3:

The authors used shRNA Ko to generate a mouse model mimicking the human movement disorders in myoclonus dystonia (DYT11). This is a very interesting and exciting paper, with extensive study of the phenotype and the beneficial effect of alcohol. In line with previous results from this team, they demonstrated that the main dysfunction lays in the cerebellum and deep cerebellar nuclei, with abnormal pattern of activity. Their results are in line with the cerebellar dysfunction observed in humans; Overall, this work sheds additional light on the human disorder and will inspire clinical research. A few issues should be addressed as below:

– As the authors intend to mimic the clinical phenotype of humans, they should be particularly careful in describing the abnormal features observed in mice;

– The term "tic-like" is inappropriate and should be avoided and the word "jerks " should be preferred, and become more precise with the word 'myoclonus". In patients with myoclonus dystonia, there are no tics (and the word tics is associated with the fact that there are repetitive, sometimes semi-purposeful movements or behaviors that can be refrained.

– From the videos, the movements are brisk and relatively short and look like jerks (they are longer in duration than those observed in myoclonus dystonia in humans (usually less than 150 ms) but, one does not expect to have a perfect copy of all the abnormal movements observed in humans dues to different motor control and behavior between species.

– Suggestion: in the videos, it seems that there are some periods when the mouse "freeze" or walks backwards (avoidance behavior), could these movements be related to behavioral disorders (as there are some in humans, one of the most pronounced being anxiety disorders).

In this case, on could imagine that modifications in the cerebellum and cerebellar networks would not only trigger movement disorders but also behavioral disorders as, more and more, the "cognitive and behavioral" functions of the cerebellum are mentioned (see recent reviews on Myoclonus dystonia).

– There is an extensive study of the electrophysiological activity in the cerebellum and the deep cerebellar nuclei. This is a great observation with "clinical" and "physiological" coherence. For further studies, could the authors explore whether cerebellar abnormal activity would influence pallidal activity (as there is an abnormal pattern of pallildal activity in human with myoclonus dystonia). This would be in line with the anatomical connectivity (as reported by P Strick' s group) and would greatly support the beneficial effect of pallidal stimulation.

– In addition, the authors may or may not discuss the behavioral changes that look alike those reported in the "anxiety" models in mice.

---

## [Author Response]

Essential revisions:1) To strengthen the causal relationship between knockdown of epsilon sarcoglycan, it would be very helpful if the authors plotted individual mice in their cohort, comparing either SGCE protein levels or a surrogate marker, such as GFP expression associated with the shRNA. Some of the mice without symptoms may have had a more limited infection zone within the cerebellum, for example. It appears the individual protein levels are available, given the dots overlaying the Western blot results in Figure 1.

To explore whether the absence of symptoms is due to inefficient knockdown of *Sgce,* or whether it is due to penetrance of the phenotype, we plotted the extent of Sgce knockdown, determined by qRT-PCR, against the dystonia score recorded in a subset of animals. Consistent with previous data in an shRNA-mediated mouse model of DYT1 (Fremont et al., 2017), animals with higher levels of knockdown tended to have more severe symptoms. Despite this trend, it is worth noting that 5/20 Sgce KD CB animals had more than 50% knockdown of Sgce RNA compared to WT, but a dystonia score of less than 2. It is thus possible that incomplete penetrance of the phenotype plays a role in the variability of the symptoms observed in Sgce KD CB mice. It also needs to be acknowledged that there is not always a one to one correlation between RNA levels, and the expression level of a functional protein.

2) The authors mention that DYT11 is a dominant disorder – does that mean that humans with the disorder only carry one mutant gene? Do humans have about 50% of the protein expression, and haploinsufficiency produces disease symptoms? Or is it a toxic gain of function mutation? This issue merits some discussion in the text, and how their model may or may not replicate the human state in terms of protein.

Thank you for drawing our attention to this issue. *SGCE* is maternally imprinted in DNA and RNA samples from human blood, but the imprinting pattern in the human brain is unknown. Consequently, the vast majority of patients inherit the disorder from their father, while the disorder is maternally inherited in approximately 5-10% of cases (Asmus et al., 2002; Grabowski et al., 2003; Muller et al., 2002). In mice, there is no expression of maternally-inherited *Sgce* (Yokoi et al., 2005). To reflect the inheritance pattern observed in humans, mouse models of DYT11 were generated by crossing male heterozygous *Sgce* knockout mice with wild-type females. We appreciate that our model does not reflect the complete loss-of-function of *SGCE* that occurs in humans. However, we believe that the effectiveness of ethanol on the symptoms of these mice, and conversely the lack of an ethanol effect on a model of DYT1 generated using the same approach, supports the utility of our shRNA model of DYT11.

3) The hyperactivity phenotype alluded to in the text (for the basal ganglia knockdown mice) is very interesting. Can the authors please include this data (it says "data not shown"). Quantitation of average velocity, rearing, or other hyperactivity phenotypes could be helpful. Video of each session could also be analyzed for time spent in the center versus periphery of the chamber, as this represents a phenotype linked to anxiety manipulations in mice, and could implicate the basal ganglia in the nonmotor aspects of the syndrome (as the authors suggest in the discussion).

We appreciate the reviewers’ desire for the behavioral phenotype of sgce KD BG mice to be examined more quantitatively. We analyzed a number of different parameters in these mice, and that data are now included in Figure 2, and its associated supplemental figures. Examination of the movement tracks in the open field revealed the greatest difference between sgce KD BG and NT BG mice. Example tracks, as well as the average, are now included in the manuscript. While the tracks of the animals are clearly different, this qualitative difference does not translate to a statistically significant differences in the average time spent in the center vs. the periphery (Figure 2—figure supplement 1), a parameter suggested by the reviewers for further scrutiny. This could be because this measure may not be an appropriate parameter to quantify the obvious differences in the behavior of these mice. Or, quite possibly, our ability to detect differences in the center vs periphery analysis is likely limited by the relatively small size of the Open Field chamber that we used in our experiments. Lastly, individual mouse to mouse variability may have reduced our statistical power.

4) The fact that ethanol relieves some of the motor symptoms in the cerebellar knockdown mice is compelling. However, it is not clear on why the dystonia score is noted in the original description of the mice (Figure 1), but the disability score and spinning score are noted in the ethanol experiments (Figure 3). Could the dystonia score also be plotted in Figure 3? I note that Dystonia score is displayed for the experiments using tor1A knockdown mice, so it would be best to include the same outcome measure for SGCE mice. Were saline injections done as controls for the effects of time and injection/handling itself on SGCE knockdown mice in the open field? The inclusion of open field distance or velocity with saline and ethanol would greatly strengthen the conclusions from these experiments.

The Disability and Spinning scores are used because we believe these scales more fully capture the range of symptoms in these mice, compared to say the Dystonia score. Nonetheless, we concur with the request to also examine the Dystonia Score, as well as the open field distance or velocity. To address these comments, the Dystonia Score in response to ethanol injection was analyzed and is now presented in Figure 3. The Dystonia score is significantly reduced in these mice after demonstration of ethanol. Despite significant reductions in the dystonic symptoms, there was no consistent difference in the distance traveled by the mice after ethanol (Figure 3—figure supplement 1). We have also included control experiments with mice injected with saline (Figure 3—figure supplement 2). There was no difference in the Dystonia score or distance traveled after injections of saline.

5) The single-unit recordings from cerebellar neurons are interesting vis a vis the circuit mechanism of dystonia. Though the animals were head-fixed, were the authors able to note any dystonic posturing during the recordings? And if so, did the firing vary during dystonic episodes? This might help get at whether this is a fixed physiological abnormality and/or there are acute changes in cerebellar firing during dystonic movements. Likewise, did they note relationships between the severity of the motor phenotype in the open field and the degree of physiological abnormality noted during the single-unit recordings?

While sgce KD CB mice exhibit dystonia, we did not perform EMG recordings in agonist and antagonist muscles at the time of single unit recording. We were thus unable to determine whether dystonic episodes were time-locked to high frequency burst firing. Despite this ambiguity in the moment-to-moment state of the animal during electrophysiological recordings, the increased ISI CV observed in Purkinje cells and DCN neurons in sgce KD CB mice were calculated over extended periods of time and reflect a combination of transient and persistent abnormal motor activity in these animals.

[Editors' note: further revisions were requested prior to acceptance, as described below.]

Reviewer #3:The authors used shRNA Ko to generate a mouse model mimicking the human movement disorders in myoclonus dystonia (DYT11). This is a very interesting and exciting paper, with extensive study of the phenotype and the beneficial effect of alcohol. In line with previous results from this team, they demonstrated that the main dysfunction lays in the cerebellum and deep cerebellar nuclei, with abnormal pattern of activity. Their results are in line with the cerebellar dysfunction observed in humans; Overall, this work sheds additional light on the human disorder and will inspire clinical research. A few issues should be addressed as below:– As the authors intend to mimic the clinical phenotype of humans, they should be particularly careful in describing the abnormal features observed in mice;– The term "tic-like" is inappropriate and should be avoided and the word "jerks " should be preferred, and become more precise with the word 'myoclonus". In patients with myoclonus dystonia, there are no tics (and the word tics is associated with the fact that there are repetitive, sometimes semi-purposeful movements or behaviors that can be refrained.– From the videos, the movements are brisk and relatively short and look like jerks (they are longer in duration than those observed in myoclonus dystonia in humans (usually less than 150 ms) but, one does not expect to have a perfect copy of all the abnormal movements observed in humans dues to different motor control and behavior between species.

We thank the reviewers for his or her comments. Throughout the text, we have changed the description from “tic-like”, which may be associated with a distinct clinical symptom in humans, to “jerking.”

– Suggestion: in the videos, it seems that there are some periods when the mouse "freeze" or walks backwards (avoidance behavior), could these movements be related to behavioral disorders (as there are some in humans, one of the most pronounced being anxiety disorders).In this case, on could imagine that modifications in the cerebellum and cerebellar networks would not only trigger movement disorders but also behavioral disorders as, more and more, the "cognitive and behavioral" functions of the cerebellum are mentioned (see recent reviews on Myoclonus dystonia).

We thank the reviewer for his or her observations and feedback. Walking backwards is something that we have observed in several of our mouse models of both ataxia and dystonia, so we did not initially consider it as a specific sign of a behavioral disorder. With that said, to follow the suggestion made by the reviewer we manually scored “walking backward” in our mice This was very time-consuming and the reason for our slow turnaround of the paper. As can be seen in Author response image 1, there seemed to be substantial increase in backward walking with sgce KD in the cerebellum. However, walking backwards was most often observed with persistent dystonic postures. Therefore, we feel that it might not be prudent to conclude that increased walking backwards was due to cerebellar effects on anxiety-like behavior and not due to cerebellar effects on movement. As such, we have not included this figure in the manuscript.

**Author response image 1. respfig1:** Increased frequency of walking backward observed in sgce KD CB mice. Mice were recorded for 5 minutes in the open field. Walking backward was scored manually by 1 observer blind to the condition of the animal. Increased walking backward was observed in sgce KD CB, but not NT CB, sgce KD BG, or NT BG mice. 1-way ANOVA with Holm-Sidak’s correction for multiple comparisons: p < 0.0001 for WT vs. sgce KD CB, NT CB vs. sgce KD CB, sgce KD CB vs. NT BG, and sgce KD CB vs. sgce KD BG. N, WT = 12; N, NT CB = 14; N, sgce KD CB = 20; N, NT BG = 12; N, sgce KD BG = 19. All data in all figures are represented as mean together with the standard deviation.

– There is an extensive study of the electrophysiological activity in the cerebellum and the deep cerebellar nuclei. This is a great observation with "clinical" and "physiological" coherence. For further studies, could the authors explore whether cerebellar abnormal activity would influence pallidal activity (as there is an abnormal pattern of pallildal activity in human with myoclonus dystonia). This would be in line with the anatomical connectivity (as reported by P Strick' s group) and would greatly support the beneficial effect of pallidal stimulation.

We thank the reviewer for these comments. The effects of abnormal cerebellar activity on basal ganglia activity in sgce KD mice is an area of future exploration. We have made a minor edit to the Discussion to more clearly address this idea:

“Another interesting possibility, consistent with pervious work on the anatomy of cerebellar-basal ganglia connectivity, is that irregular activity of the cerebellum subsequently drives irregular activity in the striatum, leading to motor symptoms in dystonia. DBS of the GPi may relieve dystonic symptoms by disrupting the abnormal output from the basal ganglia, caused in part by cerebellar dysfunction, and thereby restoring some functionality to the system.”

– In addition, the authors may or may not discuss the behavioral changes that look alike those reported in the "anxiety" models in mice.

To further address the reviewer’s interest in anxiety-like behaviors, we examined “time spent inactive,” “supported rearing,” and “unsupported rearing.” Our ability to measure “freezing,” as the reviewer recommended, was limited by our tracking software. In an attempt to capture “freezing,” we measured duration of time spent at an "Activity level" below 1.00% of all activity (Author response image 2). “Activity Level” measures changes in all the pixels of the video.

**Author response image 2. respfig2:** Cumulative duration of time spent inactive in sgce KD CB and sgce BG mice did not differ from NT CB or NT BG mice, respectively, or WT mice. 1-way ANOVA with Holm-Sidak’s correction for multiple comparisons: p > 0.05 for WT vs. NT CB, WT vs. NT BG, WT vs. sgce KD CB, WT vs. sgce KD BG, NT CB vs. sgce KD CB, and NT BG vs. sgce KD BG; N, WT = 12; N, NT CB = 14; N, sgce KD CB = 20; N, NT BG = 12; N, sgce KD BG = 19.

Rearing behavior has also been suggested previously as a way to measure anxiety-like behavior in our mice. We examined supported rearing, or rearing against the walls of the chamber, and unsupported rearing, or rearing without touching the walls of the chamber, in our mice (Author response image 3).

**Author response image 3. respfig3:** Rearing is decreased in sgce KD animals. (**A**) Supported rearing was reduced in sgce KD CB animals, compared to WT and NT CB mice. 1-way ANOVA with Holm-Sidak’s correction for multiple comparisons: p < 0.0001 for WT vs. sgce KD CB and NT CB vs. sgce KD CB; N, WT = 12; N, NT CB = 14; N, sgce KD CB = 20; N, NT BG = 12; N, sgce KD BG = 19. (**B**) Supported rearing was reduced in both sgce KD CB and sgce KD BG animals compared to WT and NT CB, and WT and NT BG mice, respectively. 1-way ANOVA with Holm-Sidak’s correction for multiple comparisons: p = 0.0086 for WT vs. sgce KD CB; p = 0.0071 for NT CB vs. sgce KD CB; p < 0.0391 for WT vs. sgce KD BG; p < 0.0001 for NT BG vs. agce KD BG; N, WT = 12; N, NT CB = 14; N, sgce KD CB = 20; N, NT BG = 12; N, sgce KD BG = 19.

Both supported rearing and unsupported rearing were reduced in sgce KD CB mice. This is likely due to the deficit in movement observed in these mice. In contrast, only unsupported rearing was reduced in sgce KD BG mice. Unsupported rearing is a measure that is sensitive to stress and context{Sturman, 2018 #952}, and a reduction in unsupported rearing, but not supported rearing, in sgce KD BG mice may reflect increased anxiety in these animals (Sturman et al., 2018).

**References**

1)Sturman, O., P.L. Germain, and J. Bohacek, Exploratory rearing: a context- and stress-sensitive behavior recorded in the open-field test. *Stress*, 2018. 21(5): p. 443-452.